# Impacts of the COVID-19 pandemic on food prices: Evidence from storable and perishable commodities in India

**Subir Bairagi**[1]*, **Ashok K. Mishra**[2], **Khondoker A. Mottaleb**[3]

**1** Department of Agricultural Economics and Agribusiness, University of Arkansas, Fayetteville, Arkansas, United States of America, **2** Morrison School of Agribusiness, Arizona State University, Mesa, Arizona, United States of America, **3** International Maize and Wheat Improvement Center (CIMMYT), El Batán, Texcoco, México

* skbairag@uark.edu

**Data Availability Statement:** The data can be found at https://microdata.worldbank.org/index.php/catalog/3830.

**Funding:** This study was supported by the Marley Foundation Fund

## Abstract

The supply chain disruptions caused by the COVID-19 outbreak have led to changes in food prices globally. The impact of COVID-19 on the price of essential and perishable food items in developing and emerging economies has been lacking. Using a recent phone survey by the World Bank, this study examines the impact of the COVID-19 pandemic on the prices of the three essential food items in India. The results indicate that price of basic food items such as *atta* (wheat flour) and rice increased significantly during the pandemic compared to the pre-pandemic period. In contrast, during the same period, the price of onions declined significantly. The findings may suggest panic-buying, hoarding, and storability of food items. The results further reveal that remittance income and cash transfers from the government negatively affected commodity prices. Thus, this study's findings suggest that families may have shifted the demand away from essential foods during the pandemic.

## Introduction

The deep and prolonged shocks of COVID-19 that we are experiencing have not been seen since the Great Depression. This time is different in terms of global effects (such as disruptions to the food supply chain and economic recession), historically low oil prices, and high commodity stocks [1, 2]. More importantly, producer and consumer prices have been moving in opposite directions [3], leading to a higher price spread than a normal economic condition [4], which is contrary to the findings of the vast literature on commodity price transmission [5]. While the price transmission literature suggests varying speed of price adjustment in the vertical supply chain, agricultural prices co-exist (producer and consumer prices move together) with a higher price at the retail level [5]. We argue that producer prices in many developing countries like India are likely to decline due to supply chain disruptions caused by the COVID-19 pandemic and the lack of storable infrastructure for perishable commodities. On the other hand, consumer prices are likely to increase due to hoarding of storable commodities influenced by consumers' psychological factors of panic buying [6–9] and the rent-seeking

**Competing interests:** The authors have declared that no competing interests exist.

behavior of processors or wholesalers [10]. Since addressing all of these above questions is challenging, the present study investigates only the impact of COVID-19 induced lockdown on the prices of three essential food items (wheat flour, rice, and onions) of Indian households.

Economic theory suggests that the equilibrium price of a commodity changes due to changes in demand and supply. The COVID-19 induced movement restrictions and lockdowns have battered both supply and demand sides of food markets. However, the degree of perturbation of the equilibrium price depends on a country's macroeconomic conditions, economic structure, and culture. More importantly, price changes may depend on whether a food is an essential or luxury item, an imported or locally produced, and a perishable or storable commodity. For instance, Torero [11] noted a differential effect of the COVID-19 induced turmoil on prices of storable and perishable items, such as wheat and rice. The author noted that the wheat and rice prices jumped by 8% and 25%, respectively, because of the world's stockpiles. In Myanmar, rice export prices increased significantly due to COVID-19 than domestically produced rice prices [12]. In India, Narayanan and Saha [13] observed that during the COVID-19 lockdown, the prices of pulses, edible oils, and perishable commodities like potatoes and tomatoes increased significantly compared to the pre-lockdown period.

Similarly, Imai, Kaicker, and Gaiha [9] found that rice, onion, potato, and tomato prices in three states of India (Maharashtra, Jharkhand, and Meghalaya) increased during the lockdown period. On the contrary, Ceballos, Kannan, Kramer [14] noted the price of tomatoes decreased significantly after the lockdown policy in Haryana. Prices of fruits and vegetables were also falling during the lockdown period compared to the pre-lockdown period in various states of India (Jharkhand, Assam, Andra Pradesh, Karnataka, Jammu, and Kashmir, [15, 16]). In the case of wheat producer price in Haryana, the effects of COVID-19 was negligible [14]. In response to the resurgence of COVID-19, no significant changes in the wholesale prices of storable staple foods (rice and wheat flour) [17] were found in China. However, prices of Chinese cabbage and fresh fruits and vegetables increased significantly in China [17–19]. A summary of the effects of COVID-19 on food prices in Asia is presented in S1 Table (see Supplementary Material).

The above studies shed some light on the impact of COVID-19 on food commodities (fresh, perishable, and storable) in large economies like India and China at the farm gate or wholesale level or at the aggregate level (district, state, or country level). However, most of the above studies fail to address the impact of COVID-19 on retail prices of essential food items such as wheat flour (*atta*), rice, and onions at the consumer or household level. Note that Indian households consume daily food items like wheat flour (*atta*), rice, and onions in some form or another. Importantly, India is the second-largest rice, wheat, and onion producer globally [20]. To put this into perspective, in 2019, India produced nearly 178 million tons (Mt) of paddy (rough rice), about 134 Mt of wheat, and about 23 Mt of onions (FAOSTAT, 2021). In terms of international trade, India is a net exporter of wheat, rice, and onion. However, food prices in India during COVID-19 increased significantly, even higher than observed in the global markets (Fig 1). Finally, these essential commodities are used as strategic plans by political parties in their mandates. It is evident that commodity prices in India have significantly influenced the win and loss of political parties in an election in India. Therefore, it is puzzling why food prices were increasing, even with an ample supply of essential commodities from local sources and politically strategic commodities in a largely populated country.

This study investigates the impact of COVID-19 induced lockdown on the prices of three essential food items of Indian households: wheat flour (*atta*), rice, and onions. Our study contributes to the literature in two ways. First, the study uses the World Bank's unique and recent household-level data with high-frequency phone surveys [21]. Second, the study assesses the impact of COVID-19 on retail prices of one perishable and two storable commodities for

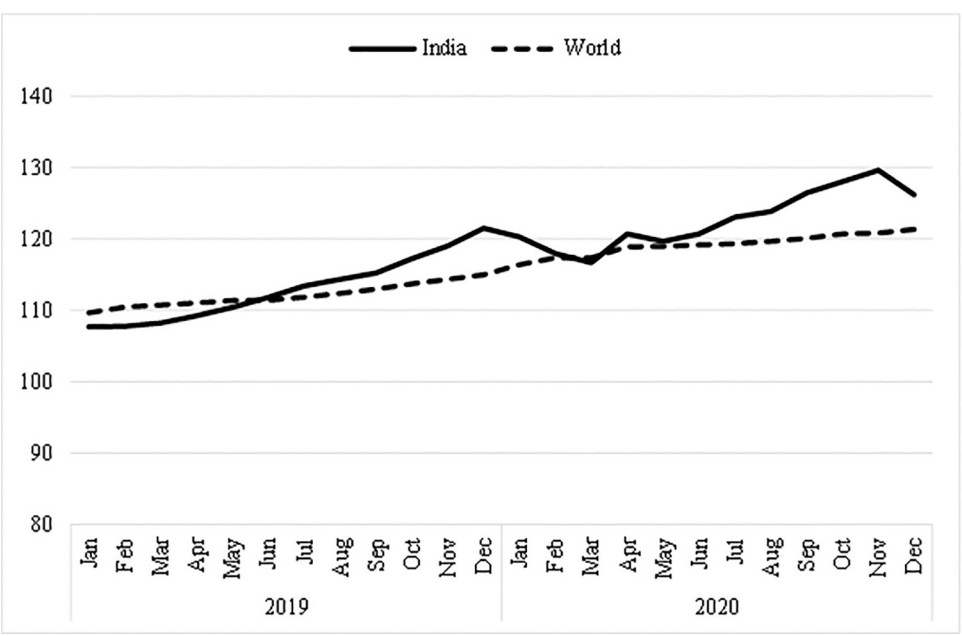

**Fig 1. Food price indices in India versus world.** Source: FAOSTAT (2021), available at http://www.fao.org/faostat/en/

consumers in a large population. We find that during the COVID-19 induced lockdown in India, prices of rice and wheat increased significantly, but onion prices decreased instead. The nature of these commodities being storable and perishable could be one of the main reasons for such price behavior. We also argue that an increase in prices of essential food items could be due to the structure and institutions of the Indian economy that are different from developed economies, such as weak market infrastructure and ineffective antitrust laws. These findings will alert policymakers and donor agencies that a COVID-19 like disruption to the economy may alter self-sufficiency programs, and domestic surplus of essential commodities might not be enough to stabilize domestic prices. Therefore, it is imperative to focus on strengthening the distribution and supply chain management systems of essential commodities in India to ensure food security in situations like the COVID-19 pandemic.

## COVID-19 and behavior of food markets in Asia: A review?

The acute respiratory syndrome coronavirus-2 (SARS-CoV-2) or COVID-19 has affected nearly every country, sickened more than 182 million, and killed more than 4 million people by this disease [22]. However, the impacts of COVID-19 cannot be judged only by the number of affected countries and people affected by this disease. Indeed, researchers still need to investigate the indirect effects such as the income and price effects of the pandemic. For instance, to control the spread of COVID19, countries across Asia, Europe, and the Americas imposed varying levels of restrictions on the movement, instituting several lockdowns, international travel bans, and social distancing measures [23, 24]. The movement restrictions and lockdowns disrupted the food supply chain, reduced production and trade, and thus significant impact on income and food security around the globe [7, 23, 25, 26].

According to the International Monetary Fund (IMF), the world gross domestic product plummeted by 4.4% in 2020 due to the COVID lockdowns and turmoil compared to the previous year [27]. As a result, around 88–115 million additional people were pushed into extreme

poverty [28]. The number of food-insecure people increased as high as 211 million (a 27.8% increase) due to the pandemic-induced food price hikes (9% increase in grain prices) [29]. Furthermore, the global maritime trade reduced approximately 7–10% during the first eight months of 2020, equivalent to US$ 225–412 billion in trade value losses [30]. Moreover, globally, the labor market participation rate in the first quarter of 2020 declined by 4.5%, equivalent to a loss of 130 million full-time jobs [31].

The adverse income effects of COVID-19 translated into food markets. For example, Nordhagen et al. [32] found that more than 94% of small food supply chains from sixteen developing countries experienced a reduction in access to inputs and financial services and decreased sales due to the COVID-19 outbreak. Nearly 45% of Chinese workers lost half of their income [33], and two in three Indian workers became unemployed during the lockdown compared to the pre-lockdown period [34]. Food prices are likely to decrease because of a negative income shock on food demand. However, food prices in many countries increased significantly, particularly at the wholesale and retail levels (see S1 Table), which may be attributable to panic buying and hoarding for staple foods [7, 35]. Either a rise in food price decreases household disposable income. As a result, their purchasing power decreases, and poor and marginalized families are likely to be affected the most. For example, in Nepal, Singh et al. [36] found that the COVID-19 pandemic caused severe food insecurity among the disadvantaged communities and low-income families that relied on daily wages and remittance income.

On the supply side factors, the disruptions in transportation sectors due to the COVID-19 lockdowns and movement restrictions severely affected food markets, creating supply shortages and increasing transaction costs [37]. For example, Ceballos, Kannan, and Kramer [14] observed that although production, harvest, and transportation costs increased for tomato farmers in the Haryana state of India, they received significantly lower prices than a typical year. Similarly, Bangladeshi vegetable farmers received less than 50% of the price compared to a typical year. The reasons could be the absence of storage facilities [38] and middlemen that link the forward supply chain (market access) [39]. The labor shortage was one of the main factors of India's production cost during the pandemic [13, 38]. On the retailer side, during the COVID 19 lockdown, large e-retailers like Flipkart and Amazon faced significant challenges in maintaining profitability. Indeed, Sangvikar et al. [40] recently documented various issues related to e-retail supply chains faced by Flipkart and Amazon in India and concluded that operation enhancement and delivery performance improvement are keys to cost optimization. Thus, taking the above actions could lead to firm profitability.

Various food trade and fiscal policies imposed to reduce the impact of COVID-19 could have also affected food markets in several ways. For instance, since the onset of the COVID-19 pandemic, more than 20 countries around the globe have imposed export bans on several essential agri-food products [41]. This trade policy restriction may have increased food prices in the world market [42], increased the risk of food insecurity in many low- and middle-income countries [43]. Finally, the government's COVID-19 stimulus programs and interventions may have led to inflationary pressures, thus, reducing investment in a productive sector. Khan et al. [44] noted that healthcare budgets had increased significantly to fight against COVID-19 with various stringency measures in many developing and emerging economies. As a result, public spending in other sectors and food supply was disrupted, which directly and indirectly increased undernourishment.

In sum, COVID-19 affected both supply and demand of food markets, resulting in higher food prices, leading to food insecurity and poverty in many developing and emerging countries. As mentioned above, various studies have investigated economy-wide impacts of COVID-19 induced turmoil [45, 46], effects on labor markets [23, 33, 47, 48], household income and food security [14, 26, 28, 36, 38, 49–52], and trade flows [41, 53]. However, the

literature on price effects of the COVID-19 pandemic is minimal. The present study addressed the impact of COVID-19 on retail prices of three essential food items (wheat flour, rice, and onions) at the consumer or household level in India.

## COVID-19 in India and government action

On January 30, 2020, the first case of COVID-19 in India was reported. On March 24, 2020, a country-wide lockdown was announced by Prime Minister Narendra Modi. The lockdown restrictions to the movement of people and goods were announced when farmers in several states started to harvest their crops [38]. On April 20, 2020, the lockdown policy was subsequently relaxed, beginning in an area designated as a non-hotspot. Since then, the central government has been issuing a series of guidelines paving the way for a phased re-opening of activities and gradual relaxation and removal of restrictions across the country. To lessen the effects of the lockdown policy, on May 12, the Prime Minister announced an economic relief package of around 10% of the country's Gross Domestic Product (GDP), including previously announced monetary and fiscal policies.

The impact of COVID-19 on the Indian economy has been significant and widely felt. The International Monetary Fund (IMF) estimated the country's GDP growth rate was the lowest (1.9%) for 2020 since the 1991 balance-of-payments crisis [48]. To safeguard its economy from the impact of the COVID-19 pandemic, the Indian government has coordinated various fiscal, monetary, and exchange rate policy measures. The direct-spending fiscal measures include in-kind (e.g., food and cooking gas) and cash transfers to lower-income households, insurance coverage for healthcare workers, and wage support and employment provision for low-wage workers. Other measures include providing credit support to businesses (approximately 1.9% of GDP), poor households, primarily migrants and farmers (1.6% of GDP), distressed electricity distribution companies (0.4% of GDP), and targeted support for the agricultural sector (0.7% of GDP). Approximately 0.3% of GDP was also allocated for some other various support measures. Among the business-support package, key elements are various financial sector measures for micro-, small-, and medium-sized enterprises, and non-bank financial companies. Finally, an additional relief package was aimed to provide concessional credit to farmers and a credit facility for street vendors.

One can expect that without the government relief package, the gravity of the pandemic effects would have been more aggravated. Therefore, the impact of the policy measures mentioned earlier and the severity of the COVID-19 pandemic in India is worth noting. For example, in a recent study, Laborde et al. [28] found real consumption decreased by about 3.9% in India, and agricultural-food exports declined by about 31%. Studies by Harris et al. [16] reported a supply shortage of various food items such as meats, fish, dairy, and vegetables immediately after the onset of the COVID-19 pandemic in India. To cope with the loss in consumer income and decreased food consumption, the Indian government enacted social safety programs. Several existing government procurement policies were also enacted to prevent a decline in wheat prices during the pandemic, ensuring stable income for wheat farmers. Additionally, Ceballos et al. [14] report that government procurement policies (such as minimum support price [MSP]) to secure more food for Indians resulted in less wheat export. In other words, the negative spillover effect of government policies in the international market. The authors also note that the price of tomatoes, a commodity not covered by MSP, initially increased, encompassed a higher degree of volatility, and then decreased as the lockdown continued into the summer months. Farmers flooded tomatoes in the local retail markets resulting in a downward shift in prices since it is a perishable commodity and there is a lack of farmers' cold storage facilities [54].

On the other hand, Narayanan and Saha [13] found that India's food prices in several urban areas increased significantly due to the COVID-19 lockdown. The authors compared prices between four weeks before and four weeks after lockdown and found that prices increased 3.5%, 15%, and 28% for edible oils, potatoes, and tomatoes, respectively. Indeed, the above studies provide some evidence of price increases in key food categories. However, ambiguity may persist in the results of a study when it comes to whether a storable or perishable food item (wheat vs. tomato) was analyzed, the surveys were conducted (immediately before or after lockdown vs. four weeks before and after lockdown), and what duration of the lockdown was considered (21 days, two, or three months). We contribute to the Indian COVID-19 literature by investigating the effect of COVID-19 induced lockdown on the prices of three essential food items (*atta*, rice, and onions) of Indian households from the six biggest states covering over 4,000 samples surveyed by the World Bank.

## Survey data and descriptive statistics

The study uses two rounds of high-frequency phone surveys by the World Bank. The survey was designed to seek and collect information to understand COVID-19 related shocks in rural India. Since market prices of food items were not collected in round 3, we did not include it in this study. Round 1 and round 2 were administered during May 5–10, 2020, and July 19–23, 2020, covering six of India's major states (Andhra Pradesh, Bihar, Jharkhand, Madhya Pradesh, Rajasthan, and Uttar Pradesh). The surveys were implemented with Computer Assisted Telephone Interview (CATI) software (The surveys are produced by the Data Production and Methods Unit of the Development Data Group, World Bank [21]), deploying it through surveyors' smartphones. Then surveyors called the selected respondents via mobile phones and recorded their responses. The survey used the samples and phone numbers from three prior IDinsight projects and the National Rural Livelihoods project conducted by the Ministry of Rural Development, Government of India, to choose a respondent. The surveys attempted to reach 11,738 households, but the response rates were approximately 55% and 46% for rounds 1 and 2, respectively. However, in this study, we used a sample of 4,297 families, making a panel dataset to track the same respondents in both rounds (Table 1). The details of data, sampling method, and data collection techniques can be obtained from the World Bank [21].

The survey added several modules to the questionnaire, including indicators related to agriculture, income and consumption, migration, access to financial relief, and health conditions. To examine the central objective of the present study, we used several indicators, mainly from agriculture and income and consumption modules. For instance, the main variables are food prices of essential commodities and time-variant variables such as government transfers and migration (Table 2). The rationale of using these two time-variant variables is that government

Table 1. Distribution of sample size used in the study.

| Phone survey sites | Respondents interviewed (number) | % of total |
|---|---|---|
| Rajasthan | 1,684 | 39.19 |
| Uttar Pradesh | 680 | 15.82 |
| Bihar | 203 | 4.72 |
| Jharkhand | 1,113 | 25.9 |
| Madhya Pradesh | 207 | 4.82 |
| Andhra Pradesh | 410 | 9.54 |
| Total | 4,297 | 100 |

Source: Authors' computation from the high-frequency phone surveys on COVID-19 in India.

**Table 2. Variable definition and summary statistics.**

| Variables | Definition | Mean | Standard deviation |
|---|---|---|---|
| Government supports | If received direct government transfers (= 1 if yes; 0 otherwise) | 0.48 | 0.50 |
| Migration | If any family member migrated (= 1 if yes; 0 otherwise) | 0.15 | 0.35 |
| P_Atta | Price of Atta (Wheat flour) in INR/kg. | 26.29 | 8.36 |
| P_Onion | Price of onion in INR/kg. | 40.41 | 46.71 |
| P_Rice | Price of rice in INR/kg. | 32.69 | 24.58 |
| *Location (state)* | | | |
| Rajasthan | 1 = if the respondent resides in Rajasthan, otherwise zero (base location used in the regression) | 0.39 | 0.49 |
| Uttar Pradesh | 1 = if the respondent resides in Uttar Pradesh, otherwise zero | 0.16 | 0.36 |
| Bihar | 1 = if the respondent resides in Bihar, otherwise zero | 0.05 | 0.21 |
| Jharkhand | 1 = if the respondent resides in Jharkhand, otherwise zero | 0.26 | 0.44 |
| Madhya Pradesh | 1 = if the respondent resides in Rajasthan, otherwise zero | 0.05 | 0.21 |
| Andhra Pradesh | 1 = if the respondent resides in Madhya Pradesh, otherwise zero | 0.10 | 0.29 |
| Observations | | 4,297 | |

Notes: Authors' computation from the high-frequency phone surveys on COVID-19 in India. INR stands for the Indian rupee (1 US$ = 73 INR, in 2020). Other jobs category is the base group.

transfers and migration can directly affect household income and consumption expenditure. Although the data include characteristics of household heads such as age, education, and ethnicity, we did not include them in the regression because of the time-invariability nature of these variables.

Fig 2 illustrates the prices of three essential food items (*atta*, rice, and onions) during the pre-lockdown (first week of March in 2020) and lockdown (any weeks of May in 2020) periods. We find that wheat flour (atta) and rice prices increased nearly by 4.0% and 11.3%, respectively. In contrast, onion prices decreased approximately 50% in the lockdown period

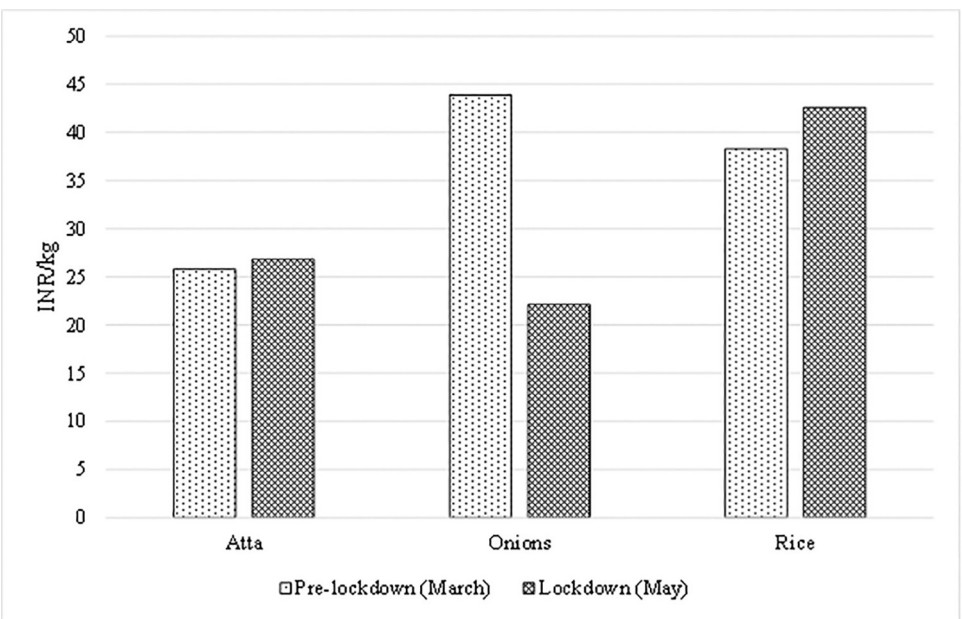

**Fig 2. Differences in essential food prices in India (before and during the lockdown in 2020).** Source: Authors' computation from the high-frequency phone surveys on COVID-19 in India.

compared with the pre-lockdown period. Since *atta* and rice are storable commodities and onions are perishable, we hypothesize that in(de)creased prices might be attributable to products' cyclical, storable, and perishable nature.

Table 2 presents two time-variant sociodemographic profiles of the sampled household heads and location variables. We find that nearly one-half of the sampled households received different forms of direct government transfers to encounter COVID-19 related shocks. Approximately 15% of the sampled households had at least a family member that is a migrant worker abroad. Finally, Table 2 reveals that samples are disproportionality collected from six states in India. Using states dummy variables will control for state heterogeneities.

## Empirical strategy

Theoretically, the equilibrium market price of a commodity is determined by its supply and demand in the market. Therefore, any factors that could affect either the supply and demand for that commodity would affect the commodity price. Previous studies suggest that the COVID-19 policy restrictions affected both the supply and demand sides of food markets worldwide [7, 17, 18, 37, 55]. Besides the lockdown, a commodity's supply and demand can be impacted by various factors, depending on a country's weather variability, cyclical nature of the product, macroeconomic conditions, economic structure, and culture. For example, Ruan et al. [17] argued that the lockdowns in China might cause the reduction of vegetable production due to reduced labor activities. However, in India, since farmers that mainly produce staple foods (rice and wheat) reside primarily in villages (houses are usually isolated from each other), crop production is unlikely to be affected due to the lockdown-induced reduced labor activities. Instead, farmers could be affected by an increase in transaction cost, mainly due to transportation costs, access to inputs, and less demand from mid-stream suppliers, such as millers and wholesalers. More importantly, market prices can be affected by food supply shortage due to the seasonality nature of a crop, such as rice is needed to grow for three to four months. However, during the pandemic, it is likely that product availability could be attributable to disruptions to the food supply chain rather than decreased production [7]. This argument can be validated by looking at India's rice and wheat production data during the pandemic year 2020—rice and wheat production increased by 2.6% and 4.1%, respectively, compared to the previous year [56]. On the demand side, as mentioned before, the COVID-19 crisis could create social panic, and consumers may hoard food in some cases. Therefore, it may lead to a vicious cycle of increasing food prices (such as Vercammen [57] and Yu et al. [18]). However, in the Indian case, since most consumers are poor, panic-buying is less likely. But food hoarding may be witnessed at the sellers' level due to the rent-seeking behavior of processors or wholesalers [7]. Finally, a product's nature being storable (e.g., rice and wheat) and perishable (e.g., onions and tomatoes) could be one of the main reasons for unexpected price behaviors of essential commodities in India.

To this end, we estimate the following reduced-form of inverse demand function [58], similar to what Ruan et al. [17] used, to estimate the effect of the COVID-19 outbreak on prices of essential food items.

$$\ln p_{it} = \beta_0 + \beta_1 T_{it} + \delta X + \gamma Z + u_{it} \qquad (1)$$

where $p_{it}$ is the price of a food item such as rice paid by $i^{th}$ household at period $t$; $T$ is an indicator variable indicating pandemic time equals one and zero pre-pandemic time; $X$ is a vector of time-variant household characteristics such as receiving government transfers and remittance income; $Z$ is a vector of location dummy variables. $\alpha$, $\beta$, $\delta$, $\gamma$, and $\theta$ are the parameters to be

estimated; $u_{it}$ captures random errors assumed to be independent and identically distributed. We estimate Eqs 1 and 2 with STATA 15.

## Results and discussion

Table 3 presents the parameter estimates of the impact of COVID-19 lockdown on the prices of two storable commodities (wheat flour and rice) and one perishable item (onion) in India. The results show that the COVID lockdown policy had a significant effect on the prices of all three commodities. In the case of *atta* (wheat flour) and rice, storable items, the lockdown significantly impacted prices. Specifically, the lockdown period increased wheat flour prices by 3% and rice prices by 16%. Perhaps this increase in price for wheat flour could be attributed to increased demand or supply shortage. An increase in demand for *atta* and rice could be for several reasons. First, *atta* and rice are essential commodities by Indian consumers. Also, depending on the consumer's location, most Indians eat either chapatis (made of *atta*) or rice in conjunction with proteins (pulses or meats) and a side of the vegetable serving. Thus, the demand for these items is unlikely to decline much due to a price change. Secondly, *atta* and rice are storable and easily transportable commodities. Therefore, prices of these foods could be increased due to supply shortage because of the rent-seeking behavior of intermediaries (hoarding) in the supply chain and disruptions to transportation sectors. Our finding is consistent with Imai, Kaicker, and Gaiha [9], who found that the COVID pandemic increased retail

**Table 3. The impact of the COVID-19 pandemic on wheat flour, rice and onions prices in India.**

| Independent variables | Prices of essential food (INR/kg), log | | |
|---|---|---|---|
| | *Wheat Flour (Atta)* | Onions | Rice (milled) |
| Pandemic time (yes = 1) | 0.030*** | -0.610*** | 0.162** |
| | (0.01) | (0.01) | (0.07) |
| Government transfer received (yes = 1) | -0.022*** | 0.002 | -0.196*** |
| | (0.01) | (0.01) | (0.07) |
| Migrated members in the family (yes = 1) | 0.020** | 0.058*** | 0.045 |
| | (0.01) | (0.02) | (0.12) |
| *Location fixed effects, base location, is Rajasthan* | | | |
| Uttar Pradesh (yes = 1) | -0.025*** | 0.242*** | Na |
| | (0.01) | (0.02) | |
| Bihar (yes = 1) | 0.068*** | 0.189*** | Na |
| | (0.01) | (0.03) | |
| Jharkhand (yes = 1) | 0.141*** | 0.249*** | Na |
| | (0.01) | (0.02) | |
| Madhya Pradesh (yes = 1) | 0.024* | 0.021 | Na |
| | (0.01) | (0.03) | |
| Andhra Pradesh (yes = 1) | na | 0.304*** | Na |
| | | (0.02) | |
| Constant | 3.200*** | 3.449*** | 3.546*** |
| | (0.01) | (0.01) | (0.07) |
| Observations | 5435 | 7231 | 694 |

Notes: Standard errors in parentheses.

* $p < .1$

** $p < .05$

*** $p < .01$. na stands for not available. INR stands for the Indian rupee (1 US$ = 73 INR, in 2020).

rice prices in Maharashtra. However, the authors found an insignificant effect of the COVID pandemic on the retail rice prices in India as a whole.

Indians prefer to purchase groceries and fresh and in-season vegetables and fruits by visiting local sellers (wet markets and farmers markets) once or twice a week. However, during the COVID lockdown, consumers could use online sellers (such as Walmart–Flipkart acquired part of Ninjacart, ShopKirana, and Jumbotail) to buy storable commodities such as *atta* and rice. Our estimated prices changes for rice and *atta* at the household level are greater than those obtained by Narayanan and Saha [13] (the average changes in *atta* and rice prices were about 0.54 and 0.98 percent, respectively). 114 Centers where the data is collected by State Civil Supplies Departments and published by the Ministry of Consumer Affairs, Food and Public Distribution, Government of India. The reader should note that Narayanan and Saha [13] did not utilize any regression model to predict the prices and did not control for regional locations of the centers when calculating price changes between pre-and post-lockdown for *atta* and rice.

Table 3 further reveals that the lockdown period had a negative and significant effect (at the 1% level of significance) on the price of onions. The lockdown period decreased onion prices by about 61%. An explanation for this result is related to the production and semi-perishable nature of onions in India. The onion is a semi-perishable crop, and around 30–40% of the harvested crop is lost during storage. The losses go beyond 40% during natural calamities, leading to heavy stress both on demand and supply. The bulbs harvested from the Rabi season have better storage life than Kharif and late Kharif onion. The crop harvested during the *Rabi* season accounts for 60% of onion production and hits the markets from March to June. The same crop meets consumer demand until October-November before the *Kharif* crop is harvested and brought to the market. Our finding is consistent with Ali and Khan [15], who compared wholesale prices of vegetables, including onions) across 12 *mandis* in Jammu and Kashmir, before and during the lockdown, and found a decrease in the wholesale price of onion. Similarly, Ceballos, Kannan, Kramer [14] found that CVOID-19 lockdown resulted in a sharp decline in farm gate prices of tomatoes in Haryana. Our finding contrasts with Imai, Kaicker and Gaiha [9] that found a positive and significant effect of the COVID pandemic on the retail price of onions and tomatoes. Similar results were reported by Çakır, Li, and Yang [19], Ruan, Cai and Jin [17], and Yu et al. [18] in China—mostly reporting higher prices of cabbage and other vegetables due to COVID-19.

Labor shortage and limited capacity of cold storage could affect a perishable product's price. The COVID lockdown presented a unique situation where warehouse owners faced labor shortages. In India, warehouses use labor to load and unload trucks, sorting and cleaning loose dry skins of the stored onions. As a result, warehouse owners may have resorted to getting the onions to the retailers—i.e., over-supply and thus lower retail prices. In addition, consumers in India prefer to purchase onions and vegetables, and fruits by visiting local sellers and farmers' markets once or twice a week. Under the COVID lockdown, consumers could not leave their homes for weekly or bi-weekly shopping resulting in an over-supply of onion in the market. Onion demand fell by 30–40 percent due to decreased demand from hotels, restaurants, and roadside eateries—closed due to the COVID lockdown. Finally, Varshney et al. [54] noted that the COVID lockdown reduced the number of intermediaries between farmgate and regional markets. Therefore, smallholders opted for local retail markets, increasing local supply and lowering market prices of perishable commodities [38].

In the econometric analysis, we also included information on government cash transfer and remittance in the family to capture the effect of additional income on *atta*, onion, and rice prices. Unfortunately, the data did not include information of the amount of government transfers or remittance income to the family. Thus, only dummy variable was included in the

regression. One can consider these variables as temporary and permanent income sources. Table 3 reveals that direct government transfers to consumers (cash transfers) and remittance income have differential effects on commodity prices during the COVID lockdown. The results indicate that families receiving government transfers are substituting away from *atta* and rice. In contrast, families receiving remittance income are buying more *atta* and onion. The above findings indicate that timely government intervention in cereals (wheat and rice), where the Indian government is a major buyer, has played a crucial role in keeping the supply chain intact and perhaps had a dampening effect on wheat flour prices and rice. Our finding is consistent with Varshney et al. [54].

Finally, the state dummy variable in Table 3 shows heterogeneity in *atta*, onion, and rice prices. Perhaps the state location dummy variables capture local market conditions, income levels, lockdown conditions, and the effectiveness of the COVID lockdown. For example, even though the Indian Prime Minister declared the COVID lockdown, states were instructed to implement lockdown at their pace and information they received on the ground. Some states like Punjab, Haryana, Bihar, Andhra Pradesh imposed a lockdown on weekends or a night curfew.

## Conclusions and policy implications

The coronavirus disease 2019 (COVID-19) outbreak continues to create economic disruptions worldwide [59]. The economic downturn resulted from the COVID-19 outbreak, and the subsequent lockdowns significantly affected labor markets and reduced trade. Millions of people worldwide lost their jobs, and incomes have dropped. For instance, nearly 45% of Chinese workers lost half of their income due to the COVID-19 crisis [33]. Two in three Indian workers became unemployed during the lockdown compared to the pre-lockdown period [34].

More importantly, the COVID-19 pandemic created a price spike in primary staple food markets [60]. For instance, the price of rice, a staple food for the globe and India, increased by 25–30% between March and May 2020 relative to the same period in 2019. Despite a good production year in 2020 and high commodity stocks, product availability in the market fell, which could be due to increased demand from panic-buying [7, 59, 61]. Surprisingly, food prices in India increased at a higher rate than what was observed in the world. A recent global study noted that COVID-19-induced changes in income and food prices could alter global food security [29]. The authors estimated that approximately between 136 and 271 million people in 72 low and middle-income countries could be food insecure due to the income effects of the outbreak. However, the price effects of the outbreak at a country-level are limited. The present study examines the impact of the COVID-19 pandemic on the prices of the three essential food items (rice, *atta*, and onions) in India.

The results revealed a differential impact of an unexpected disaster in the price of essential food commodities in a developing country such as India. For instance, the price of essential food items like *atta* (wheat flour) and rice increased significantly (3% and 16%, respectively) during the pandemic compared to the pre-pandemic period. In contrast, during the same period, the price of onions declined significantly (-61%). The result of the unforeseen disaster on prices could depend on the availability, storability, and crop cycle. We also suspect that hoarding and panic-buying could be other reasons for a price spike, which are very common in many developing countries like India (such as Dawe [62], Mahajan and Tomar [7], Pandey et al. [63]). Therefore, the government of India should take necessary policy actions to stabilize food prices, such as supplying more staple foods from the public food distribution system into the market during a crisis. Indeed, as Khan et al. (2021) argued, governments in developing and emerging economies can formulate policies that strengthen collaborations among entities in the food supply chain. Importantly, a strong and effective policy frame is needed to prevent

sellers from increasing food prices during any pandemic. Finally, the results also show that remittance income and cash transfers from the government negatively affected prices. Thus, this study's findings suggest that families may have shifted the demand away from essential foods during the pandemic. Perhaps, this finding can be related to various economic relief packages that the government offered for needy families. However, extending and strengthening distribution programs are likely to be needed to avoid leakage in the system. We argue that our findings will alert policymakers and donor agencies that a domestic surplus of essential commodities might not be enough to stabilize prices during an economic shock such as COVID-19. Therefore, to ensure food security in situations like the COVID-19 crisis, particularly in developing countries, it is imperative to focus on the essential commodities' distribution and supply chain management system.

## Limitations

Despite insightful and thought-provoking results, this research is not without limitations. Two limitations are worth noting: we considered a few essential commodities (three) and a smaller sample, particularly for estimating rice price. Future studies could use more recent data with various food items and newer estimation techniques. Also, considering the entire supply chain, identifying the factors of erratic price behavior during a crisis would provide more insights to policymakers to design an effective price stabilization policy. Finally, since India is heterogeneous in different tastes and preferences, future studies could examine consumption data (price and quantities) of more durable and non-durable food items.

## Supporting information

**S1 Table. The effects of the COVID-19 pandemic on food prices in Asian economies: Results from previous studies.**
(DOCX)

## Acknowledgments

We thank the editor and four anonymous reviewers for their constructive comments that helped improve the manuscript. The findings and conclusions in this article are those of the authors and do not necessarily reflect the views of the authors' affiliated institutions. All errors and omissions are the responsibility of the authors.

## Author Contributions

**Conceptualization:** Subir Bairagi.

**Data curation:** Subir Bairagi.

**Formal analysis:** Subir Bairagi.

**Investigation:** Subir Bairagi.

**Methodology:** Subir Bairagi.

**Software:** Subir Bairagi.

**Validation:** Subir Bairagi, Ashok K. Mishra, Khondoker A. Mottaleb.

**Visualization:** Subir Bairagi.

**Writing – original draft:** Subir Bairagi, Ashok K. Mishra, Khondoker A. Mottaleb.

**Writing – review & editing:** Subir Bairagi, Ashok K. Mishra, Khondoker A. Mottaleb.

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
