## [Decision Letter · Decision Letter 0]

11 Oct 2021

PONE-D-21-29286Impacts of the COVID-19 pandemic on food prices: Evidence from storable and perishable commodities in IndiaPLOS ONE

Dear Dr. Bairagi,

Thank you for submitting your manuscript to PLOS ONE. After careful consideration, we feel that it has merit but does not fully meet PLOS ONE’s publication criteria as it currently stands. Therefore, we invite you to submit a revised version of the manuscript that addresses the points raised during the review process.

We look forward to receiving your revised manuscript.

Kind regards,

László Vasa, PhD

Academic Editor

PLOS ONE

Journal Requirements:

Reviewers' comments:

Reviewer's Responses to Questions

**Comments to the Author**

1. Is the manuscript technically sound, and do the data support the conclusions?

Reviewer #1: Yes

Reviewer #2: Partly

Reviewer #3: Yes

Reviewer #4: Partly

2. Has the statistical analysis been performed appropriately and rigorously? 

Reviewer #1: Yes

Reviewer #2: N/A

Reviewer #3: Yes

Reviewer #4: I Don't Know

3. Have the authors made all data underlying the findings in their manuscript fully available?

Reviewer #1: No

Reviewer #2: Yes

Reviewer #3: Yes

Reviewer #4: No

4. Is the manuscript presented in an intelligible fashion and written in standard English?

Reviewer #1: Yes

Reviewer #2: No

Reviewer #3: Yes

Reviewer #4: Yes

5. Review Comments to the Author

Reviewer #1: I liked your paper: you chose a signficant topic, and wrote in a very logical and comprehensive manner.

I would only recommend to indicate at the end of the manuscript the limitations of the study (i.e. that transfers and remittances were not included in the model), or giving hints for further research possibilities.

Reviewer #2: Dear Author,

While reading your work, I found the topic interesting but at the same time, I find some lack of proper design and limited methodology. As such, I want to draw the following issues:

1- The literature gap and the novelty of this work are not evident and clear;

2- There is a lack of data analysis and synthesis of the results;

3- Methodology used is very simple and does not offer a complete picture of what really happened t o the consumption of the three categories you present.

I hope you will take into consideration these comments and improve your work accordingly.

Best wishes!

Reviewer #3: The article discusses an interesting topic given the impact of the covid crisis on the price of staple foods in developing countries. The current covid crisis has affected food security in developing countries, as a result of dysfunctions in supply chains. The authors motivated well the choice of the three foods for their study considering their share in the consumption of the local population but also India's position on the international market

The article has the potential to be published but needs to be restructured and improved.

My main recommendations are:

1. The authors could emphasize, in the introduction, the covid crisis relationship - food security and SDGs (SDG 2 - zero hunger).

2. The introductory section is too long, I propose to break it down into two sections Introduction and Literature review.

3. In the Literature review section, the presented studies could be divided into several categories considering the complexity of the analyzed phenomenon –for example the impact of the covid crisis on supply chains, the impact of the covid crisis on population behavior (compulsive shopping, herd behavior) etc. For this section, additional refences coul be used

a) Alam, G. M., & Khatun, M. N. (2021). Impact of COVID-19 on vegetable supply chain and food security: Empirical evidence from Bangladesh. Plos one, 16(3), e0248120.

b) Di Crosta, A., Ceccato, I., Marchetti, D., La Malva, P., Maiella, R., Cannito, L., ... & Di Domenico, A. (2021). Psychological factors and consumer behavior during the COVID-19 pandemic. PloS one, 16(8), e0256095.

c) Loxton, M., Truskett, R., Scarf, B., Sindone, L., Baldry, G., & Zhao, Y. (2020). Consumer behaviour during crises: preliminary research on how coronavirus has manifested consumer panic buying, herd mentality, changing discretionary spending and the role of the media in influencing behaviour. Journal of risk and financial management, 13(8), 166.

4. In the conclusions section, the authors must present the limits of the research (for example, the small number of products analyzed) and the future directions of research.

5. The theoretical and practical implications of the study need to be extended. Please, see for example …………..Khan, S. A. R., Razzaq, A., Yu, Z., Shah, A., Sharif, A., & Janjua, L. (2021). Disruption in food supply chain and undernourishment challenges: An empirical study in the context of Asian countries. Socio-Economic Planning Sciences, 101033.

6. The population was affected differently by this black swan event, the vulnerable categories having to be in the attention of the public authorities. Specific measures should be proposed and implemented by public authorities to reduce the impact of the covid crisis on vulnerable groups. Please, see for example

a) Singh, D. R., Sunuwar, D. R., Shah, S. K., Sah, L. K., Karki, K., & Sah, R. K. (2021). Food insecurity during COVID-19 pandemic: A genuine concern for people from disadvantaged community and low-income families in Province 2 of Nepal. Plos one, 16(7), e0254954.

7. In the discussion section, the authors must present the results of similar studies that confirm or refute the conclusions of the analysis performed by them.

8. Certain phrases need to be reworded (certain words are repeated or the authors have not found the best wording For example line 325-326 ”Therefore, governments in India could take necessary actions such as distributing staple foods from its public food distribution system”

Reviewer #4: Quality of paper is average in its current state. Do brainstorming, read more and conceptualise a clear idea.

Indians do not only eat aata, Onion and Rice. India being the largest producer of food grains and with large storages of food grains, there was no scarcity of food.

Refer: https://www.indiatoday.in/india/story/in-6-years-over-40-000-tonnes-of-food-grains-damaged-in-fci-godowns-1696650-2020-07-03

https://www.indexmundi.com/agriculture/?country=in&commodity=wheat&graph=production

Wheat and Rice is not the only consumable by Indians. Indians do consume Pulses, oil seeds, Fruits, vegetables, Wheat, Jawar, Bajra and other types of millets.

Though suppychain was a challenge, it did not have burden on supply side. Hording by end users, wholesalers and retailers had impact. Government policies had positive impact.

While talking about ‘walmart, flipkart’ and ‘, smallholders opted for local retail markets’ you missed avenue supermarts and reliance retail.

Structure and Flow of paper should be: Introduction, Structured Literature Review, Research Questions / Hypothesis, Research Methodology, Analysis of Data and Discussion, Findings and Conclusion. Research questions must arise after Literature review.

How this study of ‘COVID-19 pandemic on food prices: 2 Evidence from storable and perishable commodities in India’ case will be of interest for international readers of the journal?

There is no useful research output in this paper. There are generalised statements and parameters.

I conclude that paper do not contribute significantly to the body of knowledge or create any significant new knowledge to justify.

Paper must be properly formatted and rewritten.

Please verify all calculations and eliminate errors if any.

I suggest major revision.

Implement changes and resubmit paper.

Best Wishes.

6. PLOS authors have the option to publish the peer review history of their article (what does this mean?). If published, this will include your full peer review and any attached files.

Reviewer #1: No

Reviewer #2: No

Reviewer #3: No

Reviewer #4: No

---

## [Author Response · Author response to Decision Letter 0]

5 Nov 2021

See the attached "response_letter"

---

## [Decision Letter · Decision Letter 1]

1 Dec 2021

PONE-D-21-29286R1Impacts of the COVID-19 pandemic on food prices: Evidence from storable and perishable commodities in IndiaPLOS ONE

Dear Dr. Baigari,

Thank you for submitting your manuscript to PLOS ONE. After careful consideration, we feel that it has merit but does not fully meet PLOS ONE’s publication criteria as it currently stands. Therefore, we invite you to submit a revised version of the manuscript that addresses the points raised during the review process.

We look forward to receiving your revised manuscript.

Kind regards,

László Vasa, PhD

Academic Editor

PLOS ONE

Journal Requirements:

Reviewers' comments:

Reviewer's Responses to Questions

**Comments to the Author**

1. If the authors have adequately addressed your comments raised in a previous round of review and you feel that this manuscript is now acceptable for publication, you may indicate that here to bypass the “Comments to the Author” section, enter your conflict of interest statement in the “Confidential to Editor” section, and submit your "Accept" recommendation.

Reviewer #2: (No Response)

Reviewer #3: All comments have been addressed

Reviewer #4: (No Response)

2. Is the manuscript technically sound, and do the data support the conclusions?

Reviewer #2: (No Response)

Reviewer #3: Yes

Reviewer #4: Partly

3. Has the statistical analysis been performed appropriately and rigorously? 

Reviewer #2: (No Response)

Reviewer #3: Yes

Reviewer #4: Yes

4. Have the authors made all data underlying the findings in their manuscript fully available?

Reviewer #2: (No Response)

Reviewer #3: Yes

Reviewer #4: Yes

5. Is the manuscript presented in an intelligible fashion and written in standard English?

Reviewer #2: (No Response)

Reviewer #3: Yes

Reviewer #4: Yes

6. Review Comments to the Author

Reviewer #2: (No Response)

Reviewer #3: The article has been improved by the authors and it can be published considering the quality of the authors' scientific approach.

Reviewer #4: I appreciate efforts taken by authors to revise paper as per comments of several reviewers. Still please try to implement few further changes. Structure and Flow of paper should be: Introduction, Structured Literature Review, Research Questions / Hypothesis, Research Methodology, Analysis of Data and Discussion, Findings and Conclusion. Research questions must arise after Literature review.

Please verify all calculations and eliminate errors if any.

I ask you to refer following articles for making literature review strong:

https://doi.org/10.1371/journal.pone.0256921 , https://doi.org/10.1504/ijenm.2021.118057 , https://doi.org/10.1371/journal.pone.0254954

Implement changes and resubmit paper.

7. PLOS authors have the option to publish the peer review history of their article (what does this mean?). If published, this will include your full peer review and any attached files.

Reviewer #2: No

Reviewer #3: No

Reviewer #4: No

---

## [Author Response · Author response to Decision Letter 1]

7 Dec 2021

PONE-D-21-29286R1

Title: Impacts of the COVID-19 pandemic on food prices: Evidence from storable and perishable commodities in India

Response to Reviewer 4

Comment: I appreciate efforts taken by authors to revise paper as per comments of several reviewers. Still please try to implement few further changes. Structure and Flow of paper should be: Introduction, Structured Literature Review, Research Questions / Hypothesis, Research Methodology, Analysis of Data and Discussion, Findings and Conclusion. Research questions must arise after Literature review.

Response: Done. We have followed the paper structure that is compatible with economic analysis papers in PLOSONE

Comment: Please verify all calculations and eliminate errors if any.

Response: In response to your comments, we have checked our estimates, and all are in good shape. 

Comment: I ask you to refer following articles for making literature review strong:

https://doi.org/10.1371/journal.pone.0256921 , https://doi.org/10.1504/ijenm.2021.118057 , https://doi.org/10.1371/journal.pone.0254954

Response: We have added these articles in the literature review section of the revised paper. However, we did not include Huang’s study in our paper. This paper is focused on West Virginia, USA, which is outside our focal areas of Asia. 

Comment: Implement changes and resubmit paper.

Response: Done

---

## [Editor Report · Decision Letter 2]

9 Feb 2022

Impacts of the COVID-19 pandemic on food prices: Evidence from storable and perishable commodities in India

PONE-D-21-29286R2

Dear Dr. Bairagi,

We’re pleased to inform you that your manuscript has been judged scientifically suitable for publication and will be formally accepted for publication once it meets all outstanding technical requirements.

Kind regards,

László Vasa, PhD

Academic Editor

PLOS ONE
---

## [Editor Report · Acceptance letter]

23 Feb 2022

PONE-D-21-29286R2 

Impacts of the COVID-19 pandemic on food prices: Evidence from storable and perishable commodities in India 

Dear Dr. Bairagi:

I'm pleased to inform you that your manuscript has been deemed suitable for publication in PLOS ONE. Congratulations! Your manuscript is now with our production department. 

Kind regards, 

on behalf of

Prof. Dr. László Vasa 

Academic Editor

PLOS ONE